# Short H2A histone variants are expressed in cancer

Guo-Liang Chew [1], Marie Bleakley[2], Robert K. Bradley [3,4,5], Harmit S. Malik[4,6], Steven Henikoff [4,6], Antoine Molaro [4,7✉] & Jay Sarthy [4✉]

Short H2A (sH2A) histone variants are primarily expressed in the testes of placental mammals. Their incorporation into chromatin is associated with nucleosome destabilization and modulation of alternate splicing. Here, we show that sH2As innately possess features similar to recurrent oncohistone mutations associated with nucleosome instability. Through analyses of existing cancer genomics datasets, we find aberrant sH2A upregulation in a broad array of cancers, which manifest splicing patterns consistent with global nucleosome destabilization. We posit that short H2As are a class of "ready-made" oncohistones, whose inappropriate expression contributes to chromatin dysfunction in cancer.

[1] The Cancer Science Institute of Singapore, National University of Singapore, Singapore, Singapore. [2] Clinical Research Division, Fred Hutchinson Cancer Research Center, Seattle, WA, USA. [3] Computational Biology Program, Public Health Sciences Division, Fred Hutchinson Cancer Research Center, Seattle, WA, USA. [4] Basic Sciences Division, Fred Hutchinson Cancer Research Center, Seattle, WA, USA. [5] Department of Genome Sciences, University of Washington, Seattle, WA, USA. [6] Howard Hughes Medical Institute, Fred Hutchinson Cancer Research Center, Seattle, WA, USA. [7] Present address: Genetics, Reproduction and Development (GReD) Institute, Université Clermont Auvergne, Clermont-Ferrand, France. ✉email: antoine.molaro@uca.fr; jsarthy@fredhutch.org

Nucleosomes, the fundamental subunit of chromatin, consist of octamers of histones (H2A, H2B, H3, and H4) that wrap 147 bp of DNA[1]. Single-allele mutations in histones, termed "oncohistones", are found in many different malignancies[2,3]. Oncohistones comprise small percentages of the total histone pool[4,5] and rarely cause cancer by themselves[2,6]. Instead, they synergize with other oncogenes to facilitate the development of neoplastic chromatin landscapes[2,6]. Recent large-scale cancer genome analyses identified recurrent mutations in histones within the highly conserved histone fold domain (HFD) in many common cancers[2,3]. These HFDs mutations, including the well-characterized H2B-E76K substitution, reduce nucleosome stability in vitro and perturb chromatin in vivo[2,7]. Similarly, most H2A HFD oncohistone mutations disrupt either contact sites with DNA (R29Q) or inter-nucleosomal interactions (acidic patch) (Fig. 1a). Cells co-expressing H2B-E76K and a PI3KCA oncogene showed increased transformation capacity, consistent with nucleosome instability, enhancing the cancerogenic potential of other oncogenes[2].

Short histone H2A variants (sH2A) are a class of histone variants expressed during mammalian spermatogenesis[8–11]. Regulation of sH2A expression in normal testis is unknown[12]. Unlike other histone variants, sH2As are rapidly evolving and possess highly divergent HFDs, mutated acidic patches, and truncated C-termini, all of which impact nucleosome stability[8,13–16]. The best characterized of these variants, H2A.B, forms unique nucleosomes that wrap ~120 bp of DNA both in vitro and in vivo[15–17]. In testis, H2A.B is incorporated into nucleosomes during meiosis and has been shown to interact with splicing factors at actively transcribed genes[17–20]. Germline disruption of H2A.B-encoding genes in mice revealed that H2A.B loss is associated with chromatin dysfunction and splicing changes in testis[18].

Though a role for sH2As in cancer has yet to be determined[21], the emerging literature on nucleosome instability as cancer driver[2,3] along with H2A.B's potent ability to destabilize nucleosomes[13–16] prompted us to investigate whether sH2As may contribute to cancer. Previous work showed that expression of H2A.B causes increased sensitivity to DNA damaging agents, shortens S-phase[22], and alters splicing[17,19,22], each of which are associated oncogenesis. Additional evidence for a role for H2A.B in cancer comes from Hodgkin's lymphoma (HL), where H2A.B transcripts have been detected[23] and HL cells expressing H2A.B grow faster than H2A.B-negative cells[22]. Here, through comparative analyses of germline short H2A sequences and oncohistone mutations in canonical H2A, we show that short H2As inherently possess oncohistone features. We explore several cancer data sets and find H2A.B expression in a diverse array of malignancies. We also show that many of these cancers possess unique splicing signatures. We propose that the nucleosome-destabilizing characteristics sH2As evolved for their role in testis result in oncohistone activity in other tissues.

## Results

**sH2As have evolved oncohistone features.** There are five X-linked sH2A genes in humans: H2A.B.1.1 (*H2AFB2*), H2A.B.1.2 (*H2AFB3*), H2A.B.2 (*H2AFB1*), H2A.P (*HYPM*), and H2A.Q (unannotated)[8]. We compared the amino-acid sequences of sH2As to canonical H2A to assess whether their rapid evolution resulted in oncohistone-like changes. This analysis revealed that many of the most common cancer-associated mutations in canonical H2A are already present in all wild-type sH2A sequences (Fig. 1a, Supplementary Fig. 1a, b). This includes R29Q/F substitutions that correspond to the second most frequent mutation in canonical H2A (Fig. 1a, Supplementary Fig. 1a, b)[2,3]. In addition, all wild-type sH2As have a C-terminal truncation that removes E121, the most common mutation in canonical H2A (Fig. 1a, Supplementary Fig. 1a, b)[2,3]. Phylogenetic analyses in primates showed that despite their rapid evolution, these oncohistone-like changes are highly conserved (Fig. 1b, Supplementary Fig. 1a–c)[8]. This conservation implies functional consequences as many of these residues are critical contact points for histone-DNA or histone-histone interactions[1,13–16]. These data

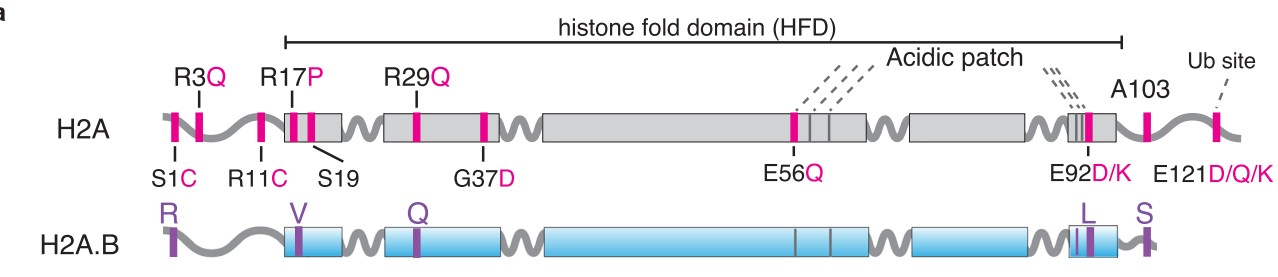

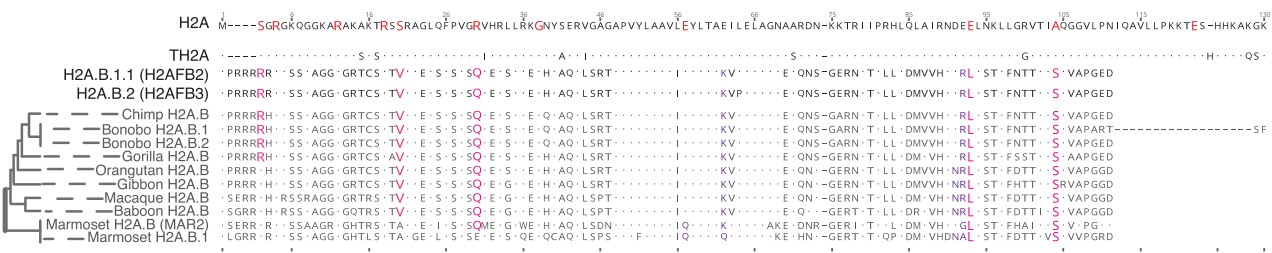

**Fig. 1 H2A.B possesses oncohistone features that are conserved throughout primates. a** Schematic of common oncomutations found in human core H2A and their status in H2A.B. Marked sites on core H2A show WT amino acid position followed by its most common cancer-specific substitution in TCGA (pink). Associated sites found in WT short H2As are shown in purple. **b** Protein alignment of core H2A, testis-specific H2A (TH2A), and H2A.B paralogs from Human and representative primates. Substitutions corresponding to oncohistone mutations in H2A (see Fig. 1.) are shown in pink.

show that sH2As contain oncohistone features similar to canonical H2A mutations in cancers.

**H2A.Bs are reactivated in a broad array of cancers.** The oncohistone properties inherent in sH2As indicate that they may play a role in cancer simply through upregulation. We focused on the expression of H2A.B paralogs, as they are well annotated and have been shown to impact both nucleosome stability and cell cycle progression[22]. To investigate whether H2A.Bs are reactivated in different cancers, we first used transcriptomic data from The Cancer Genome Atlas (TCGA). This analysis showed that H2A.B paralogs are activated (at a threshold of >1.5 transcripts per million (TPM)) in numerous individual tumors across cancer types (Fig. 2a, Supplementary Data 1, Supplementary Data 2), but never in adjacent normal tissue (Supplementary Data 1), and very rarely (<1.5%) in non-testes tissue samples from the Genotype-

Tissue Expression database (Supplementary Table 1). The range of expression varies widely, with H2A.B-encoding transcripts present at >100 TPMs in two specimens (Supplementary Data 2). Although many tumors reactivate *H2AFB1* alone, most tumors that express *H2AFB2* also express *H2AFB3* (Fig. 2a). This finding may result from transcriptional co-regulation due to their genomic proximity (Supplementary Fig. 2b) or inability to distinguish these near-identical paralogs by short-read mapping[8]. Despite their similarity, we were able to distinguish these two genes in a few tumor samples (Fig. 2a).

Across the TCGA data set, diffuse large B-cell lymphomas (DLBCLs) showed the highest frequency of aberrant H2A.B expression at 50% (Fig. 2a). A recent analysis of DLBCL genomes identified five distinct molecular subtypes[24], including a favorable prognosis-germinal center (FP-GC) subtype associated with histone mutations. We investigated whether H2A.B expression

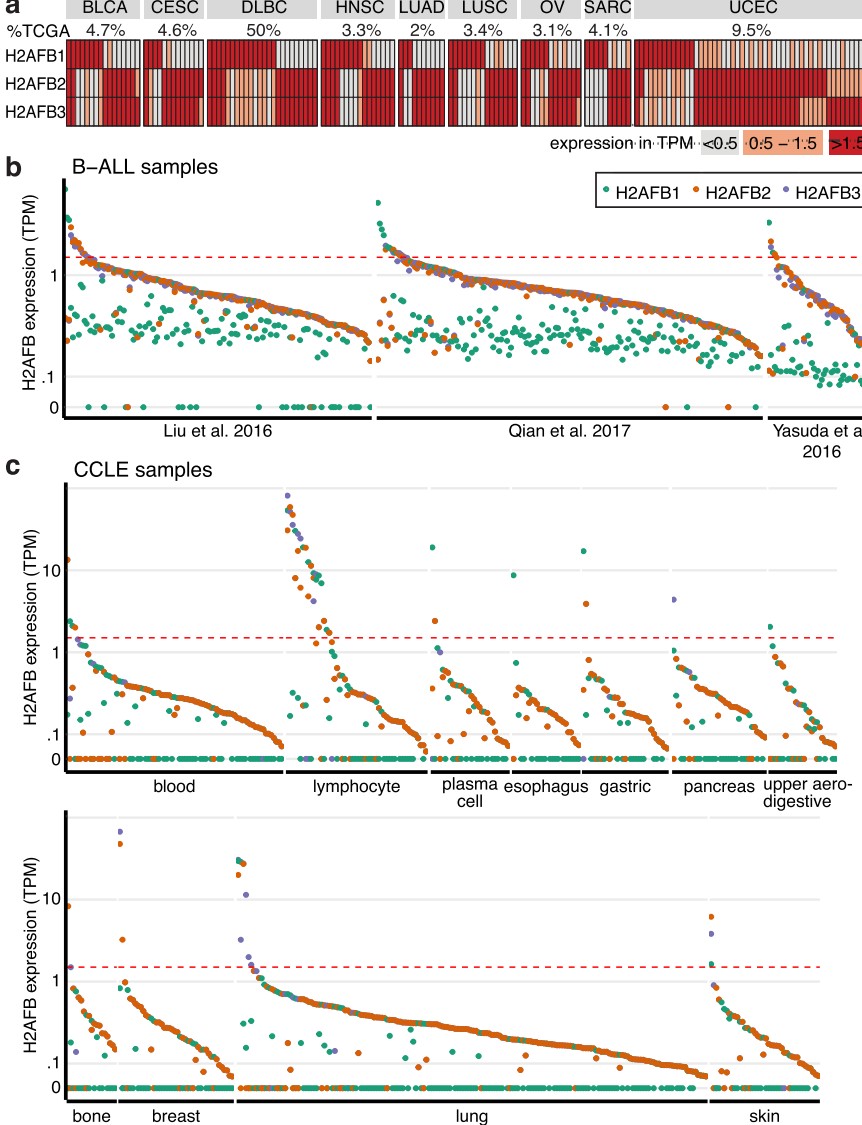

**Fig. 2 H2A.B is expressed in a broad array of cancers. a** Heat map illustrating the co-expression of H2A.B paralogues in individual tumors that express any one H2A.B paralogue (at >1.5 TPM as measured by RNA-seq), for cancer types with at least 10 tumors expressing any H2A.B paralogue. Percentages of tumors for each cancer type that express any H2A.B paralogue are shown. **b** Expression levels (TPM) of H2A.B-encoding transcripts in three independent B-acute lymphoblastic leukemia data sets, horizontal line demarcates 1.5 TPM as measured by RNA-seq. Only samples with non-zero expression of any H2A.B paralog are shown: 18, 5, and 1 samples are omitted from Liu et al. 2016, Qian et al. 2017, and Yasuda et al. 2016, respectively. **c** As in **b**, but for cancer cell lines from CCLE, grouped by their lineage. Only lineages with any sample >1.5 TPM, and samples with non-zero expression of any H2A.B paralog are shown.

was restricted to the FP-GC subtype. We queried the 37 DLBCL samples for mutations associated with the FP-GC subtype including linker H1 and core histones, immune evasion genes, PI3K, NF-κB, and JAK/STAT/RAS pathway components[24]. Twenty-five samples had a mutation in at least one of these genes, including 15 different samples with histone mutations (Supplementary Data 3). H2A.B was expressed in 13 samples with any FP-GC mutation, and in 6 of the 10 FP-GC samples without histone mutations (Supplementary Data 3). To contrast this with another DLBCL subtype, we analyzed H2A.B expression in the poorer prognosis-germinal center subtype associated with mutations in chromatin modifiers *EZH2, CREBBP, EP300, KMT2D,* and *BCL11A*[24]. Though we did not identify any *EZH2* mutations, 15 samples had mutations in at least one chromatin modifier gene. Nine of these samples also had H2A.B upregulation. These analyses suggest that H2A.B expression occurs in multiple germinal center DLBCL subtypes.

Other cancers in the TCGA data set with H2A.B aberrant expression include uterine corpus endometrial carcinomas (UCEC) (9.5%), urothelial bladder carcinomas (BLCA) (4.7%), and cervical squamous cell carcinomas and endocervical carcinomas (4.5%) (Fig. 2a). These same cancers were previously identified as having the highest frequencies of core histone mutations in the TCGA data set, ranging from 5 to 8%[2]. We found a few specimens with both recurrent H2A mutations and H2A.B expression (Supplementary Data 4), however, the low numbers of specimens that share both of these features hinder meaningful correlative analyses.

Upregulation of H2A.B in HL[23] and DLBCLs (Fig. 2a) prompted us to analyze data sets from other lymphoid lineage-derived, low mutation cancers for aberrant H2A.B expression. We queried four separate B-acute lymphoblastic leukemia (B-ALL) data sets and found 6–7% of specimens with H2A.B-encoding transcripts at >1.5 TPM (Fig. 2b) in three of the data sets[25–27], and 13% in the fourth (Supplementary Fig. 2c)[28]. Because of the diversity of liquid and solid cancers with H2A.B expression, we searched the Cancer Cell Line Encyclopedia (CCLE) database[29] for cell lines with H2A.B expression at >1.5 TPM. Consistent with high-frequency H2A.B expression in TCGA DLBCLs, lymphomas demonstrated the highest percentage of H2A.B-positive cell lines (Fig. 2c), with 70% of HL and 25% of non-Hodgkin's lymphoma cell lines expressing H2A.B. The spectrum of H2A.B expression across other cancers was also similar between CCLE and TCGA data sets (Fig. 2c). We conclude that H2A.B is aberrantly expressed in a broad array of cancers.

We investigated the potential causes of H2A.B induction in cancer. Although little is known about the transcriptional regulation of H2A.B-encoding loci in testis, changes in X-chromosome ploidy are associated with increased fitness in cancer cells[30]. We investigated whether H2A.B expression in cancer may result from global derepression of large domains, amplifications, or gain of an additional X chromosome in these samples. We compared levels of X- to autosome-linked transcripts in H2A.B-expressing and silent samples and found no significant differences (Supplementary Fig. 2a). We also investigated the expression profiles of individual *H2AFB* loci and their surrounding regions and found that upregulation was limited to each individual H2A.B-encoding locus without upregulation of neighboring loci (Supplementary Fig. 2b). These results are consistent with our findings in the TCGA data set, where median H2A.B expression for the 232 H2A.B-positive samples is ~3 TPM (Supplementary Data 2), corresponding to 49th percentile of all expressed genes. This level of expression is more likely the result of local, specific activation of individual *H2AFB* paralogues than recurrent amplifications or broader X-chromosome dysfunction.

**H2A.Bs are associated with cancer-specific, rather than pan-cancer gene expression programs**. H2A.B proteins encoded by *H2AFB1* and *H2AFB2/3* are nearly identical in sequence. Nevertheless, the independent reactivation of *H2AFB1* and *H2AFB2/3* in different cancer specimens raised the possibility that these closely related paralogues may be associated with distinct global gene expression programs. To explore this, we compared transcriptomes from *H2AFB1*-reactivated samples versus those from *H2AFB2/3*-reactivated samples within the same cancer types. We found thousands of genes that were commonly up- or downregulated in UCEC, HNSC, LUSC, and BLCA (Fig. 3a), suggesting that different H2A.B paralogues operate in similar gene expression contexts.

We investigated whether expression of other genes was consistently associated with H2A.B expression. We found 146 genes were upregulated and 90 downregulated across H2A.B-positive cancers (Supplementary Data 5). We did not identify co-upregulation of other testis-specific histone variants such as H2A.1 (TH2A) or H2B.1 (TH2B)[20] (Supplementary Data 5) in H2A.B-expressing cancers. Three histone variants with broad tissue distributions, H2A.Z, H2A.X, and H3.3, also did not show consistent differences between H2A.B-positive and negative cancers, except for lower H2A.Z and H2A.X in UCEC (Fig. 3b). We note that median H2A.X levels are similar to maximum values for H2A.B (Fig. 3b, Supplementary Data 1). We also examined expression of the histone chaperone NAP1 (*NAP1L1*), which can assemble H2A.B-containing nucleosomes[15,21]. We detected NAP1-encoding transcripts in all TCGA cancers, with DLBCLs expressing the highest levels (Fig. 3b, Supplementary Table 2). The chromatin consequences of this correlation, i.e., whether higher NAP1 levels result in increased incorporation of H2A.B in chromatin are unknown.

We noted that 12/146 of the commonly upregulated genes are Cancer-Testis Antigens. As *H2AFB1* was previously shown to be co-expressed with a subset of CTAs in HL[23], we determined whether H2A.B-reactivated cancers are generally associated with CTA upregulation. We summarized the expression of individual CTAs into a composite "CTA score" for each tumor and compared scores between H2A.B-reactivated and silent samples (Fig. 3c). Although H2A.B-expressing HNSCs, LUSCs, and UCECs showed statistically significant CTA enrichment, DLBCLs and SARCs did not (Fig. 3c). We also examined the four B-ALL data sets and found that H2A.B expression was associated with CTA upregulation (Fig. 3c). However, individual CTAs such as NY-ESO-1 (*CTAG1B*) and *CT45A5* were variably expressed across cancers (Supplementary Data 5), consistent with well-recognized transcriptional heterogeneity of this class of genes[31,32]. These data indicate that H2A.B expression is associated with CTA expression in several cancer types.

CTAs are subject to endogenous immunosurveillance mechanisms[23] and TCGA tumor samples are known to contain variable amounts of immune infiltrates[33]. We investigated whether H2A.B expression was associated with immune infiltrates, as this could confound our transcriptome analyses. We found that transcript levels for markers of B-cells, T-cell subsets, NK cells, monocytes, and activated macrophages did not show consistent enrichment across H2A.B-expressing tumors (Supplementary Fig. 3). In fact, UCEC displayed a statistically significant decrease in *PRF1* expression as well as several macrophage and neutrophil markers. Several sH2A-derived peptides are predicted to bind human leukocyte antigen (HLA) molecules[34,35] (Supplementary Data 6), suggesting an immunosuppressive microenvironment may contribute to sustained H2A.B expression in UCEC. The lack of excess immune infiltrates in H2A.B-positive TCGA specimens and the identification of H2A.B-positive cancer cell lines (Fig. 2c) support H2A.B upregulation in cancer cells, though a contribution from surrounding stroma in patient specimens cannot be excluded.

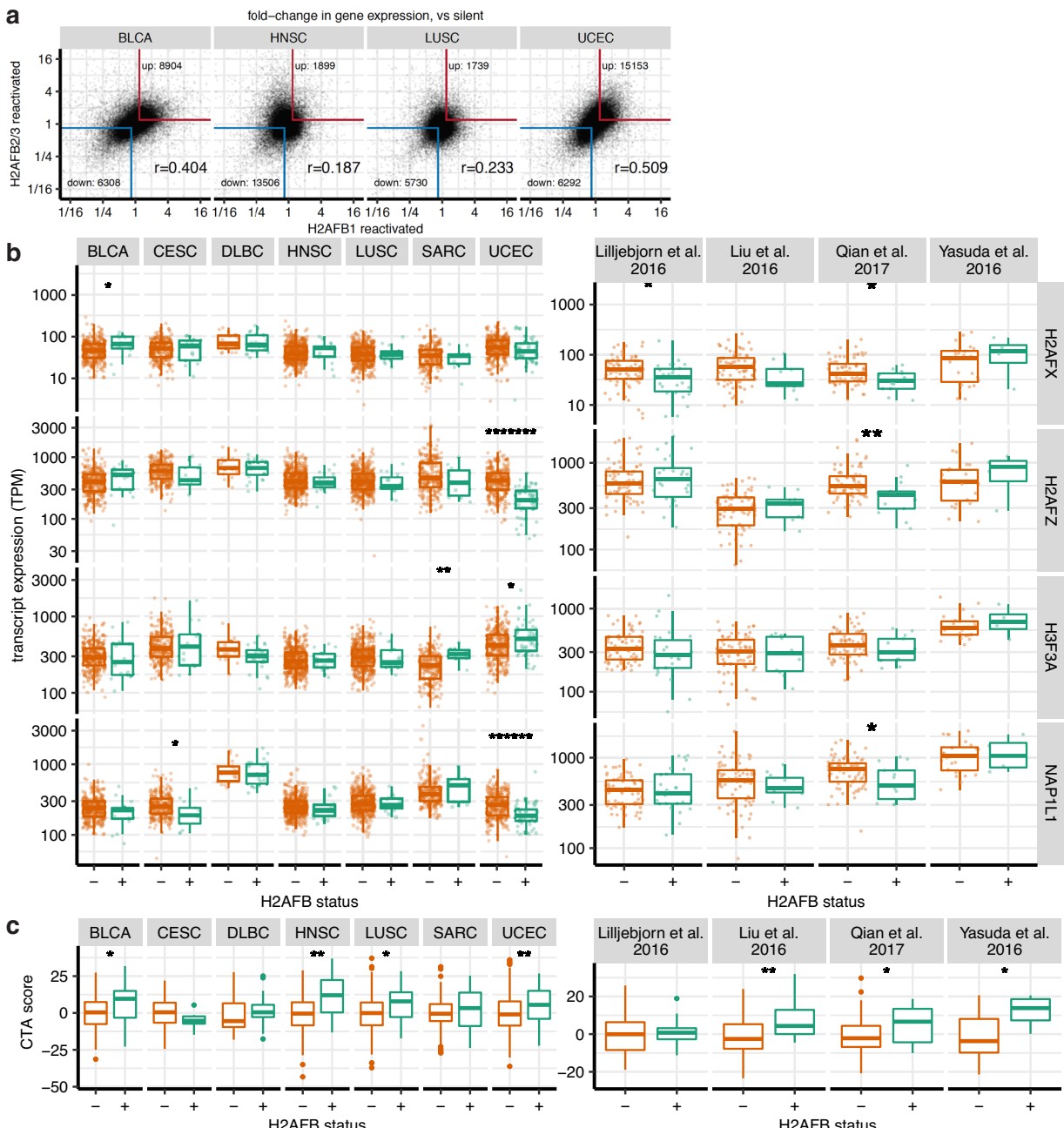

**Fig. 3 Gene expression analyses in H2A.B-reactivated cancers. a** Scatter plot of gene expression differences (expressed as fold-change), comparing *H2AFB1*-positive against negative tumors (*x* axes), and *H2AFB2/3*-positive against negative tumors (*y* axes). The red and blue borders show genes that are commonly up- or downregulated at >1.19-fold. Pearson correlation coefficient is also shown. **b** Boxplots comparing levels of H2A.Z (*H2AFZ*), H3.3 (*H3F3A*), H2A.X (*H2AFX*), and NAP1 (*NAP1L1*) transcripts in H2A.B-positive (green) vs negative (orange) cancers from TCGA and B-ALL data sets. Asterisks show the statistical significance of the difference in TPMs by a two-sided Mann–Whitney *U* test. \**p* < 0.05; \*\**p* < 0.01; \*\*\*\*\*\**p* < 0.000001; \*\*\*\*\*\*\**p* < 0.0000001. Number of cancer samples in each group are listed in Supplementary Table 2. Boxplots indicate the 1st quartile, median and 3rd quartile, whereas the whiskers extend from the box-ends to values no larger/smaller than 1.5 times of the inter-quartile range. All data points are additionally plotted. **c** Boxplots as in **b**, comparing CTAs scores in H2A.B-positive (green) and negative (orange) cancers in TCGA and B-ALL cancers. For each tumor, the expression of CTAs is summarized as a CTA score: the sum Z-normalized log expression of the top 40 most variably expressed CTAs (within each cancer type). Asterisks show the statistical significance of the difference in CTA scores by a one-sided Mann–Whitney *U* test. \**p* < 0.05; \*\**p* < 0.01. Outlier points beyond the whiskers are additionally plotted.

**H2A.B-expressing cancers have distinct splicing patterns**. H2A.B has been shown to directly bind RNA and interacts with splicing factors and H2A.B expression impacts alternative splicing patterns[17–20,22]. To determine if H2A.B expression is associated with splicing dysregulation, we annotated and quantified all constitutive and alternative splicing events in the transcriptomes of H2A.B-reactivated and silent tumors from the TCGA data set. We uncovered thousands of altered splicing events between these cancers (Fig. 4a, b). We found that H2A.B expression is associated with reduced utilization of alternative "cassette exons" (se) and proximal alternative 3′ polyadenylation (APA) sites (Supplementary Fig. 4a, b). These features were particularly prominent in

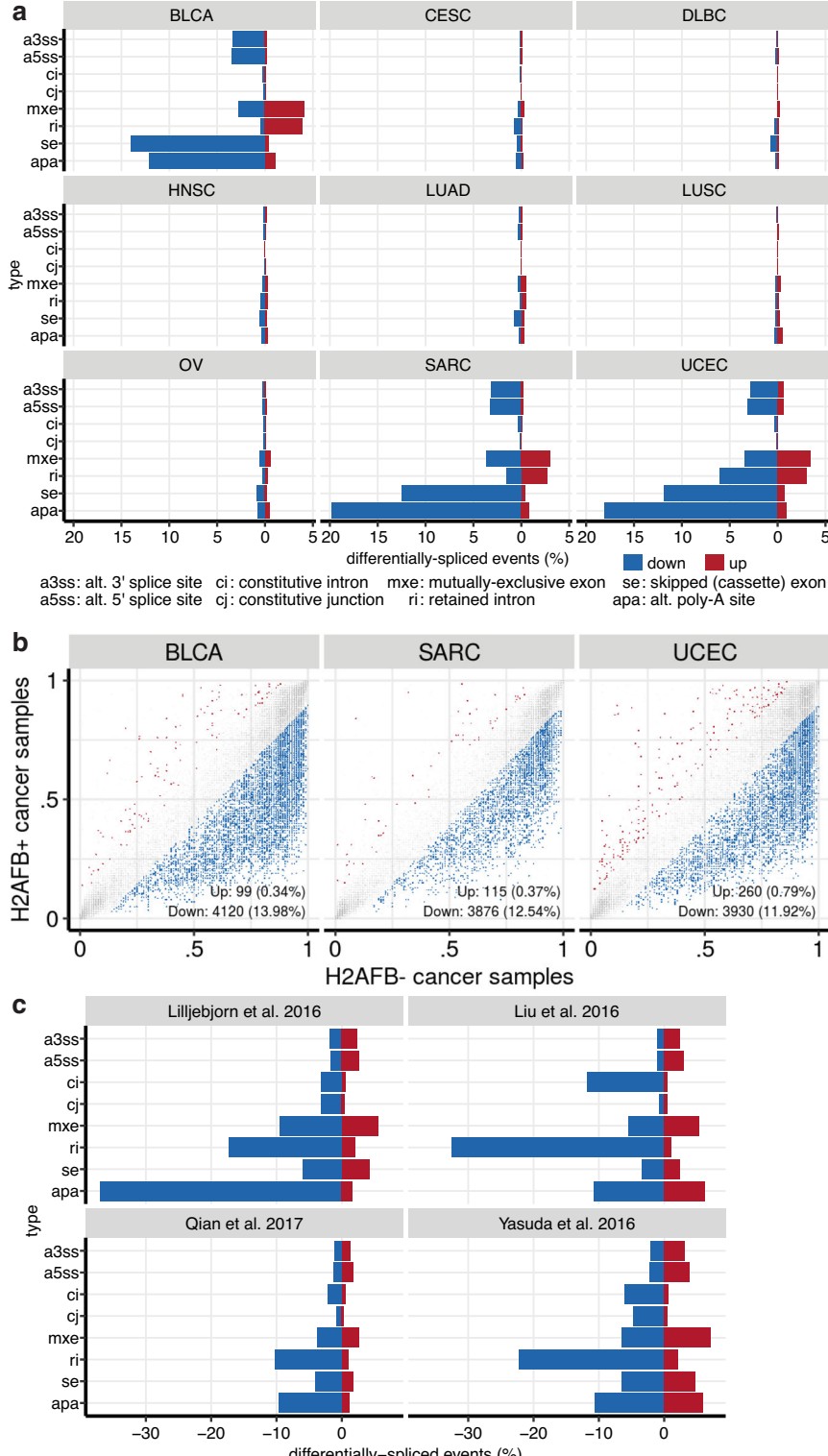

**Fig. 4 Splicing analyses in H2A.B-reactivated cancers. a** Bar graphs showing the percentage of up- and downregulated splicing events (constitutive and alternative) when comparing H2A.B-positive to negative tumors, for cancer types with at least 10 tumors expressing any H2A.B paralogue. **b** Scatter plots of alternative cassette exon inclusion for various cancers, comparing individual events from H2A.B-positive (*y* axes) to negative tumors (*x* axes). Axes units are fraction of transcripts that include the alternative cassette exon (Psi). Red and blue points indicate events that are significantly up- or downregulated (respectively) in H2A.B-positive tumors, at a threshold of $p < 0.05$ (one-sided Mann–Whitney test) and the difference in Psi >0.1. The number of significantly up- and downregulated events are tallied in the bottom of each panel. **c** As in **a**, but for B-ALL data sets.

BLCA, SARC, and UCEC (Fig. 4a, Supplementary Fig. 4a). Although the changes are individually modest (Supplementary Fig. 4c–f, Supplementary Data 7), they are widespread, i.e., we observe significant changes at thousands of sites across multiple cancer types (Fig. 4a, b). These patterns are not H2A.B paralogue-specific, as similar patterns were observed in specimens expressing either *H2AFB1* or *H2AFB2/3* (Supplementary Fig. 4a, b).

We also explored splicing in the four B-ALL data sets. Unlike in myelodysplastic syndromes and acute myelogenous leukemias[36], B-ALLs are not associated with mutations in splicing factors and global splicing dysregulation is not thought to be a major driver of these leukemias. When we compared splicing patterns in the H2A.B-reactivated and silent samples within each data set, we observed aberrant splicing at a scale similar to that seen in H2A.B-positive TCGA cancers, with reductions in alternative exon and APA usage. However, the most notable feature is a consistent decrease in retained introns "ri" in all four data sets (Fig. 4c). We conclude that H2A.B expression is associated with splicing dysfunction, with some features common among many cancers while others occur in a context-specific manner.

## Discussion

The discovery of oncohistone mutations has revealed new insights into the biology of cancer. We show that all mammalian genomes already encode sH2A histone variants that have evolved nucleosome-destabilizing features without any additional coding mutations. These features are important for sH2As' roles in normal testis physiology but result in oncohistone properties when expressed out of context. In this manner, they are similar to CATACOMB/EZHIP[37,38], another testis-specific oncohistone mimic that inhibits EZH2 in a subset of rare malignancies. Unlike CATACOMB/EZHIP, however, H2A.B expression occurs in many common cancers. The diversity of H2A.B-expressing cancer types suggests that pathological histone dynamics play a more significant role in neoplasia than previously appreciated.

The precise molecular targets of H2A.B expression in cancers are not known. Relatively few genes are commonly dysregulated across H2A.B-positive malignancies (Supplementary Data 5), implying that H2A.B impacts different genes in different cancers. As nucleosomes protect DNA from inappropriate transcription factor binding, nucleosome instability may allow oncogenic TFs access to different regulatory elements depending on cancer type[2,39]. Nucleosome destabilization also hastens RNA pol II elongation, which in turn reduces transcription-coupled splicing efficiency[40]. Alternative exons and proximal polyadenylation sequences are preferentially impacted by inefficient splicing owing to their weaker splice signals, resulting in a splicing phenotype similar to those observed in several H2A.B-positive cancers[40]. As some alternative exons promote mRNA degradation by targeting them for nonsense-mediated decay, even modest reductions in alternative splicing can increase oncogene expression[41]. H2A.B may operate at the nexus of several processes that cooperate to drive oncogenesis.

The relationship between histone dynamics, transcription, and splicing may also explain our inability to detect a splicing phenotype in DLBCLs despite high-frequency H2A.B expression. Many chromatin proteins are deranged in DLBCLs including Myc, p300, H1 linker, and core histones, each of which can also impact alternative splicing[24,42–44]. Whether potential similarities between histone mutant cancers and H2A.B-expressing cancers extend to prognoses and vulnerabilities merits further investigation, particularly in the context of DLBCL where larger data sets are needed to dissect these relationships. Several cell lines show sensitivity to *H2AFB1*-gRNAs in the Sanger Cancer Dependency

Map, with lymphoma-derived cell lines SU-DHL-8 and IM9 being among the most sensitive to *H2AFB1* disruption[45]. Better characterization of histone mutations and H2A.B expression across cancer cell lines is also needed in order to probe for similarities between H2A.B-expressing cancers and histone mutant cancers. Finally, sH2A-derived short peptides that bind HLA molecules (Supplementary Data 6) may be useful immunotherapy targets, and global splicing dysregulation can also generate highly immunogenic neoantigens[46]. Thus, our discovery of sH2A-expressing cancers may open new avenues of study and treatment for hundreds of thousands of cancer cases worldwide.

## Methods

**Alignments of sH2A sequences**. sH2A and other H2A sequences were retrieved from Histone DB v2[47] and previously published work[8]. Predicted protein sequences from annotated CDS were aligned using ClustalW and manually curated. For primate alignments, sequences were arranged according to the accepted species phylogeny.

**Genome annotation, RNA-seq read mapping, and gene and isoform expression estimation**. RNA-seq reads from TCGA were downloaded from CGHub. Reads were processed for gene expression and splice isoform ratio quantification as previously described[48]. In brief, read alignment and expression estimation were performed with RSEM v1.2.443[49], Bowtie v1.0.044[50], and TopHat v2.1.145[51], using the hg19/GRCh37 assembly of the human genome with a gene annotation that merges the UCSC knownGene gene annotation[52], Ensembl v71.1 gene annotation[53], and MISO v2.0 isoform annotation[48]. MISO v2.038 was used to quantify isoform ratios. The trimmed mean of M values method[54], as applied to coding genes, was used to normalize gene expression estimates across all of TCGA.

**Data analysis and visualization**. Data analysis was performed in the R programming environment and relied on Bioconductor[55], dplyr[56], and ggplot2[57].

**RNA-seq coverage plots**. RNA-seq coverage plots (i.e., Fig. S2c–f) were made using the ggplot2 package in R, and represent reads normalized by the number of reads mapping to all coding genes in each sample (per million).

**Somatic mutation analysis**. TCGA somatic mutation calls from the Mutect pipeline[58], were obtained using the GDCquery_Maf function from TCGAbiolinks[59]. Mutations in the following canonical histones were collated by their class: H2A—HIST1H2A(A/B/C/D/E/G/H/I/J/K/L/M), HIST2H2A(B/C), HIS3H2A; H2B —HIST1H2B(A/B/C/D/E/F/G/H/I/J/K/L/M/N/O), HIST2H2B(E/F), HIST3H2BB. Recurrent mutations (Supplementary Data 3) are defined as occurring at least five times across all cancer types in TCGA (e.g., 10 instances of E121Q mutations are found in various H2As across all TCGA samples).

**Differential gene expression and splice event analyses**. For the purposes of differential analyses, a threshold of >1.5 TPM was used to determine whether H2A. B was expressed in a sample, whereas a threshold of <0.5 TPM was used to determine whether H2A.B was not expressed; samples with an intermediate expression of H2A.B were not used in differential analyses. Statistical significance in differential expression or splicing in H2A.B-positive versus H2A.B-negative cancer samples was determined with a Mann–Whitney *U* test, as implemented in wilcox.test in R.

**Prediction of H2A variant candidate T-cell epitopes**. The amino acid sequence of human H2A.B.1.1 (*H2AFB2*), H2A.B.1.2 (*H2AFB3*), H2A.B.2 (*H2AFB1*), H2A. Q, and H2A.P (*HYPM*) were examined for short peptides with the potential to bind to common HLA alleles[34,35]. Specifically, the NetPanMHCA4.0 algorithm of the Immune Epitope Database and Analysis Resource (IEDB) was used to identify peptides of 8, 9, 10, or 11 amino acids long that are predicted to bind with strong affinity (IC50 < 300 nM) to HLA-A*0101, A*0201, A*0301, A*1101, A*2402, B*0702, B*0801, B*1501, B*1502, B*4001, B*4002, B*4402 or B*4403. Additional IEDB algorithms were employed to confirm predicted HLA binding, whereby binding was predicted by NetPanMHCBA4.0 and at least one other method (including artificial neural network, stabilized matrix method, PickPocket and NetPanMHCBA4.0) was required for inclusion in Supplementary Table 5.

**Reporting summary**. Further information on research design is available in the Nature Research Reporting Summary linked to this article.

## Data availability

All data sets used in this study are publicly available or previously published. RNA-seq reads from TCGA were downloaded from CGHub, but are now available from the NIH

NCI Genomic Data Commons [https://portal.gdc.cancer.gov/]. RNA-seq reads from B-ALL samples were obtained from the Japanese Genotype–Phenotype Archive (accession number JGAS000047, this data set is available under restricted access and step by step instructions for obtaining access including Form 2 submission are available at [https://humandbs.biosciencedbc.jp/en/data-use]. For questions regarding this data set, please contact Dr. Hiroyuki Mano at hmano@m.u-tokyo.ac.jp)[27]. RNA-seq reads from B-ALL samples were also obtained from the European Genome-phenome Archive (accession number EGAD00001002112[28], and EGAD00001002151)[26], and the Chinese Genotype–phenotype Archive (data are available under restricted access, access can be obtained by contacting Dr. Sai-Juan Chen: sjchen@stn.sh.cn)[25]. RNA-seq quantification of CCLE cell lines were obtained from the Broad Institute CCLE Portal [https://portals.broadinstitute.org/ccle/data] (02-Jan-2019 release). TCGA mutation data was obtained through the GenomicDataCommons Bioconductor package. The remaining data are available within the Article, Supplementary Information or from the authors upon request.

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

## Acknowledgements

We would like to thank R. Eisenman and members of the Malik, Henikoff, and Bradley labs for helpful discussions and comments on this manuscript. J.S. is supported by a Damon Runyon Cancer Research Foundation/Sohn Foundation Pediatric Cancer Research Fellowship, an Alex's Lemonade Stand Foundation (ALSF) Young Investigator Award and a Northwestern Mutual/ALSF Award for Data Sharing. A.M. is supported by the Damon Runyon Cancer Research Foundation (DRG:2192-14) and by the NIH (R01 GM074108). G.-L.C. is a Mahan Fellow. M.B. is supported by a Stand Up To Cancer Innovative Research Grant, Grant Number SU2C-AACR-IRG 14-17. Stand Up To Cancer is a division of the Entertainment Industry Foundation. R.K.B. is a Scholar of The Leukemia and Lymphoma Society (1344-18). H.S.M. and S.H. are investigators of the Howard Hughes Medical Institute. The results shown here are in part based upon data generated by the TCGA Research Network: https://cancergenome.nih.gov/.

## Author contributions

G.C., A.M., and J.S. designed the study and conceived the analyses. G.C., A.M., and M.B. performed the analyses. R.B., H.M., and S.H. provided guidance. A.M. and J.S. provided project leadership. G.C., A.M., and J.S. wrote the paper.

## Competing interests

The authors declare no competing interests.
