## [Peer Review File · Nature Communications]

REVIEWERS' COMMENTS

Reviewer #2 (Remarks to the Author):

In their response to Reviewer 3, I would agree with the authors' statement that their manuscript reports that (1) short H2As share features of mutant H2As observed in cancers and (2) that short H2As appear to be expressed in a subset of cancers. I don't agree with their third statement to R3 that H2A.B is expressed at levels similar to H2A.X in cancers. I think it's more fair to say, as they do in their response to me, that maximum levels of H2AB expression can achieve median levels of H2AX expression. This really is an important distinction, as median expression levels seem to be ~an order of magnitude different between H2AB and H2AX. Not to be redundant with my previous review, but this puts H2A.B median expression around 0.5% of total H2A. I think the authors are factual in the text but feel that this point is obscured. Beyond that, I don't have any additional comments and I think this manuscript is ready for publication as an "Analysis" paper should that format be available.

Reviewer #3 (Remarks to the Author):

The sequence features of these sH2A histones identified by Guo-Liang Chew et al. are the stronger part of this study. However re-expression and potential role as an oncohistone in certain tumors are not convincing:

1) Expression of these variants in tumors seems quite low and looks more like a continuum, so no clear re-activation event in tumor samples.

On the other hand, CCL4 lymphoma cell lines show a nice bimodal distribution. So, it would be of interest to analyze further the lymphoma cell lines expressing these sH2A to see if they can identify the event responsible for this higher expression in these cell lines. This could help them track back similar event in tumors.

2) Another point is that RNA level is not enough to show that this sH2A are indeed present in tumors especially viewing the relatively low RNA level they uncover (1.5 transcript per million). As the increase in expression is relatively modest, is it specific to tumor cells or is it due to neighboring cells including stromal cells? The authors should at least validate protein expression using available tumor proteomic datasets or by IHC or western blot.

3) Regarding sH2A variants specificity to subgroups of DLBCL, the authors should extend the analysis to the 3 other DLBCLs molecular subgroups and not only to 2 subtypes. It is rather unclear why they limited their analysis to only these 2 subtypes.

Overall, the data presented remains a limited set of evidence to label these sH2A histones as oncohistones. More analyses are needed to strengthen this hypothesis, including wet lab work not just only database mining analyses as is the case for this study.

REVIEWERS' COMMENTS for NCOMMS-20-37665-T

(Author responses in *italics*)

We thank the reviewers for their reviews of our transferred manuscript. We appreciate their interest in our work and believe their comments have strengthened our manuscript.

Reviewer #2 (Remarks to the Author):

In their response to Reviewer 3, I would agree with the authors' statement that their manuscript reports that (1) short H2As share features of mutant H2As observed in cancers and (2) that short H2As appear to be expressed in a subset of cancers. I don't agree with their third statement to R3 that H2A.B is expressed at levels similar to H2A.X in cancers. I think it's more fair to say, as they do in their response to me, that maximum levels of H2AB expression can achieve median levels of H2AX expression. This really is an important distinction, as median expression levels seem to be ~an order of magnitude different between H2AB and H2AX. Not to be redundant with my previous review, but this puts H2A.B median expression around 0.5% of total H2A. I think the authors are factual in the text but feel that this point is obscured. Beyond that, I don't have any additional comments and I think this manuscript is ready for publication as an "Analysis" paper should that format be available.

We completely agree this point is critical and were not trying to obscure our results. The previous version we stated that H2A.X levels are similar to maximum H2A.B levels (line 194). To be more explicit now state that "median H2A.X levels are similar to maximum values for H2A.B" (line 194-195). We are grateful to Reviewer 2 for the very helpful comments throughout the review process, they have strengthened the manuscript substantially.

Reviewer #3 (Remarks to the Author):

The sequence features of these sH2A histones identified by Guo-Liang Chew et al. are the stronger part of this study. However re-expression and potential role as an oncohistone in certain tumors are not convincing:

We appreciate Reviewer 3's recognition of important findings of our study.

1) Expression of these variants in tumors seems quite low and looks more like a continuum, so no clear re-activation event in tumor samples. On the other hand, CCLE lymphoma cell lines show a nice bimodal distribution. So, it would be of interest to analyze further the lymphoma cell lines expressing these sH2A to see if they can identify the event responsible for this higher

expression in these cell lines. This could help them track back similar event in tumors.

Expression in some TCGA cancers is actually quite high, with several specimens having >100 TPMs (Supp. Table 1). We have included a sentence highlighting this point in the text (lines 119-120).

2) Another point is that RNA level is not enough to show that this sH2A are indeed present in tumors especially viewing the relatively low RNA level they uncover (1.5 transcript per million). As the increase in expression is relatively modest, is it specific to tumor cells or is it due to neighboring cells including stromal cells? The authors should at least validate protein expression using available tumor proteomic datasets or by IHC or western blot.

In addition to showing expression of >100 TPMs in some TCGA specimens (lines 119-120), our analyses of immune infiltrates in TCGA cancers and detection of H2A.B in cancer cell lines strongly supports cancer cell-specific expression of H2A.B. We now state this in the text (lines 223-224).

3) Regarding sH2A variants specificity to subgroups of DLBCL, the authors should extend the analysis to the 3 other DLBCLs molecular subgroups and not only to 2 subtypes. It is rather unclear why they limited their analysis to only these 2 subtypes.

The TCGA DLBCL dataset is comprised primarily of germinal center DLBCLs. Our conclusion that H2A.B is expressed in multiple DLBCL subtypes is appropriate given that we detect expression in at least two different germinal center subtypes.

Overall, the data presented remains a limited set of evidence to label these sH2A histones as oncohistones. More analyses are needed to strengthen this hypothesis, including wet lab work not just only database mining analyses as is the case for this study.